# Prevalence of *Schistosoma haematobium* and Intestinal Helminth Infections among Nigerian School Children

**DOI:** 10.3390/diagnostics13040759

**Published:** 2023-02-16

**Authors:** Tolulope Alade, Thuy-Huong Ta-Tang, Sulaiman Adebayo Nassar, Akeem Abiodun Akindele, Raquel Capote-Morales, Tosin Blessing Omobami, Pedro Berzosa

**Affiliations:** 1Department of Medical Laboratory Science, Niger Delta University, Wilberforce Island 560103, Nigeria; 2Malaria and Neglected Tropical Diseases Laboratory, National Centre of Tropical Medicine, Biomedical Research Networking Center of Infectious Diseases (CIBERINFEC), Instituto de Salud Carlos III, Avenida Monforte de Lemos, 5, Pabellón 13, 28029 Madrid, Spain; 3Medical Laboratory Science Department, Ladoke Akintola University of Technology, Ogbomoso 210101, Nigeria; 4Centre for Emerging and Re-Emerging Infectious Diseases, Ladoke Akintola University of Technology, Ogbomoso 210101, Nigeria

**Keywords:** *Schistosoma mansoni*, *Schistosoma haematobium*, soil-transmitted helminths, neglected tropical diseases, Kato-Katz, Nigeria

## Abstract

Schistosomiasis and soil-transmitted helminthiases (STH) are two parasitic diseases mainly affecting school children. The purpose of this study was to estimate the current prevalence and infection intensity, in addition to the associations of these infections with age and sex, in children aged 4–17 years living in Osun State, Nigeria. From each participant (250 children), one urine and one stool sample were taken for the study, for the microscopic detection of eggs or larvae in faeces by means of the Kato–Katz method and eggs in filtrated urine. The overall prevalence of urinary schistosomiasis was 15.20%, with light infection. The intestinal helminthic species identified (and their prevalence) were *S. stercoralis* (10.80%), *S. mansoni* (8%), *A. lumbricoides* (7.20%), hookworm (1.20%), and *T. trichiura* (0.4%), all of them being classified as light infections. Single infections (67.95%) are more frequent than multiple infections (32.05%). With this study, schistosomiasis and STH are still endemic in Osun State, but with a light to moderate prevalence and light infection intensity. Urinary infection was the most prevalent, with higher prevalence in children over 10 years. The >10 years age group had the highest prevalence for all of the intestinal helminths. There were no statistically significant associations between gender and age and urogenital or intestinal parasites.

## 1. Introduction

Schistosomiasis is a parasitic disease caused by infection with *Schistosoma* spp. trematodes. The disease affects poor rural communities but has spread to urban areas and to tourists visiting endemic areas [1]. There are two main types of the disease: (i) intestinal schistosomiasis, caused by *S. mansoni*, *S. japonicum*, *S. mekongi*, *S. guineensis,* and *S. intercalatum*; and (ii) urogenital schistosomiasis, caused only by *S. haematobium* [2]. Human transmission occurs through contact with water (e.g., bathing, swimming, washing clothes in water containing the infective cercariae) infested with larval forms (cercariae) that develop in freshwater snails, the intermediate host; inadequate sanitation increases the risk of transmission [1,2,3].

As of January 2020, schistosomiasis is endemic in 78 tropical and subtropical countries, of which 51 countries have moderate to severe transmission and require preventive chemotherapy with praziquantel. Approximately 236 million are people infected worldwide, with more than 90% living in Africa, causing about 24,000 deaths in 2016 and 2.5 million disability-adjusted life years (DALYs). Deaths and DALYs are likely underestimated due to underreporting, the methods used to assess disability and other factors [4,5].

Soil-transmitted helminthiases (STH) are caused by infection with intestinal parasites (*Ascaris lumbricoides* and *Trichuris trichiura*), hookworms (*Necator americanus* and *Ancylostoma duodenale*) and roundworms (*Strongyloides stercoralis*). Human transmission occurs through eggs or larvae in faeces, which contaminate soil in areas with poor sanitation. *S. stercoralis* is transmitted similarly to other STH, requires a different diagnostic method and can cause hyper-infection syndrome leading to death [4,6,7].

In 2019, 92 countries were endemic for *A. lumbricoides*, *T. trichiura*, and hookworm, and required mass drug administration (MDA) with albendazole or mebendazol. Approximately 1.5 billion people estimated to be infected with STH, about 6300 deaths and 3.5 million DALYs in 2016. The burden of *S. stercoralis* should be quantified precisely [4,5].

In Nigeria, only two *Schistosoma* species cause human schistosomiasis, *S. mansoni,* which causes intestinal schistosomiasis, and *S. haematobium*, which causes urinary schistosomiasis. In Nigeria, schistosomiasis is a disease of considerable and growing importance, mainly affecting rural areas and vulnerable age groups. School children are the major victims of this disease [8].

Nigeria is among the countries with the highest burden of STH disease in Africa. STH mainly affects children, causing anemia, Vitamin A deficiency, malnutrition, loss of appetite, retarded growth, reduced ability to learn, etc., in them. The prevalence of this disease is moderate (<50%)–high (≥50%).

The World Health Organization (WHO) classifies schistosomiasis and STH as neglected tropical diseases (NTDs), but not *S. stercoralis* [5,9]. These NTDs in Nigeria are targeted to for their control, elimination, and eradication by the Federal Ministry of Health and Government of Nigeria, in collaboration with various stakeholders and partners. The National Schistosomiasis Control Programme was initiated in 1988, and the goal of the programme is to control/eliminate schistosomiasis in the region through MDA, delivering regular praziquantel tablets, donated by Merck KGaA Germany since 2009, to at least 75% of school-age children in endemic areas in the country in line with WHO recommendation. The STH control programme was initiated in 2007. In line with WHO recommendations, the programme has set a target of regular administration of mebendazole tablets, donation from Johnson and Johnson, to at least 75% of school-age children in endemic areas in the country at risk of morbidity. Regarding strongyloidiasis, in those areas where MDA with ivermectin has been used to control onchocerciasis or lymphatic filariasis, the prevalence of strongyloidiasis seems to have reduced, but further investigation is needed.

The purpose of this study was to estimate the current prevalence and infection intensity of *Schistosoma* and intestinal parasitic infections in a group of school children (aged 4–17 years) living in a recognized endemic area at Ore community in Odo-Otin Local Government of Osun State, Nigeria. We also evaluated the associations between the acquisition of these infections and age and sex.

## 2. Materials and Methods

### 2.1. The Study Area

This study was carried out in the Ore community in Odo-Otin Local Government of Osun State, Nigeria (Figure 1). Osun State has an estimated population of 4,275,526 people (Nigeria Population Census 2006). The community is in stable malaria transmission zone. Malaria is present during all months of the year, with a marked increase in the wet season, which normally runs from April to October. The soil in the communities can be described as well drained, moderately leached, and with moderate humus content. Farming and petty trading are the major occupations. The study was executed from September 2021 to December 2021, which spanned the dry and rainy months.

### 2.2. Ethical Considerations

Ethical clearance for the study was given by the Research and Ethics Committee of Osun State, Ministry of Health, Osogbo, Nigeria (ref no: OSHREC/PRS/569T/131). Before sample collection, meetings were held with community leaders, teachers, and community members. The aim of the study, the study procedures, types of specimens required, and benefits of the study to individuals and to the community as a whole and risks involved were included in the informed consent letters and fully explained to parents and children. Parents and legal guardians were asked to give their verbal consent for the children who were willing to be sampled after being given proper information about the study. The ethical committee allowed the use of oral consent because majority of the parents were not educated.

### 2.3. Study Population and Sampling

School children (one primary and one secondary school) were recruited into the study. A total of 250 children aged between 4 and 17 years old were selected and from each participant, one urine sample and one stool sample were taken for *S. haematobium*, *S. mansoni*, *S. stercoralis*, *A. lumbricoides*, *A. duodenale* (hookworm), and *T. trichiura* studies, using microscopic detection of eggs in faeces by means of the Kato–Katz (KK) method and eggs in urine using urine filtration. Both stool and urine samples were collected separately from each pupil using two sterile, leak-proof, and transparent wide-mouthed containers. The sample containers were pre-labelled with the participant’s identification number. The collected samples were transported within 2 h of collection to the laboratory for analysis.

Furthermore, a blood sample was also taken to determine the packed cell volume (PCV). Of the three schools in the community, only two participated in the study. The association of parents of the third school refused to participate.

### 2.4. Study Design and Eligibility Criteria

This cross-sectional study was conducted on school children in the community. All of the school children who were willing to be part of the study and reside in the study area, and who had not taken anti-helminth drugs within six months before the study, were recruited into the study. Children whose parents gave consent to participate in the study and those without a severe medical condition were recruited into the study. The exclusion criteria included anyone who was too sick to participate or who could not provide informed consent or obtain it from a parent or guardian.

### 2.5. Determination of Packed Cell Volume (PCV)

Anticoagulated blood with heparin was centrifuged in a sealed capillary tube at 10,000× *g* for 5 min. The tubes were placed in the micro-haematocrit reader and children with PCV values < 31% were considered as anaemic, which was further classified as mild (21–30%), moderate (15–20%), or severe (≤15%) [10].

### 2.6. Microscopic Examination

Microscopy was performed on one KK slide per fecal sample by a skilled, well-trained research microscopist who is an expert in detecting helminth eggs. In total, 20% of prepared (previously examined) Kato–Katz slides were randomly selected for quality control and examined by a second experienced microscopist who was blinded to the previous test results.

### 2.7. Urine Filtration Analysis

The presence of *S. haematobium* eggs was assessed using the urine filtration technique, the standard for the diagnosis of urogenital schistosomiasis recommended by WHO, as previously described [11]. Briefly, 10 mL of the freshly passed urine sample was filtered through a micro-filter membrane with a pore size of 10–12 μm (MF, Whatman, NJ, USA) using a syringe. The micro-filter membrane was then carefully placed on a glass slide, mounted on a microscope, and examined using a light microscope’s low-power objective (10×). *Schistosoma* eggs were counted and recorded as the number of eggs/10 mL of urine. Infection intensity was classified as light (<50 eggs/10 mL of urine) or heavy (≥50 eggs/10 mL of urine), as previously described by Atalabi et al. [12].

### 2.8. Detection and Quantification of Intestinal Helminths

The KK thick smear technique, the standard method for STH according to the WHO, was used for the quantitative determination of helminth ova on one KK slide per sample. The intensity of infection was expressed as the number of eggs per gram (epg) of faeces. The number of helminth eggs were counted and multiplied by 24 in order to quantify the number of epg of faeces. To ensure consistency of the result and as a form of quality control, 20% of the slides were randomly selected and read again. The epg was classified according to the WHO classification as light infection (epg < 100), moderate infection (epg 100–399), and heavy infection (epg ≥ 400) [13].

### 2.9. Statistical Analysis

Data obtained from microscopic examinations were entered in Microsoft Excel prior to statistical analyses (mean, range, percentage, and estimated prevalence) for urinary schistosomiasis and intestinal helminths, calculated using the free software WinEpi: Working in Epidemiology [14]. The confidence intervals (CI) were established at 95%. Graphs and tables were created with Microsoft Excel. Results were compared and associations between qualitative variables were determined using a Chi-square test (χ^2^), included in the free software WinEpi. *p*-values were also calculated, considering *p*-values < 0.05 to be statistically significant.

## 3. Results

### 3.1. Study Population

A total of 250 children aged 4–17 years were screened using microscopy test for schistosomiasis and STH evaluation. The mean age of the 250 recruited children was 9.72 years, 135 out of 250 were boys (54%) and 115 out of 250 were girls (46%). The mean age for the boys was 9.91 and that for the girls was 9.50. Children were distributed in three age groups: <5 years (2/250, 0.8%); 5–10 years (157/250, 62.8%); >10 years (91/250, 36.4%).

### 3.2. Packed Cell Volume and Anaemia

From the 250 children tested for anaemia, according to PCV values, 99 (39.6%) were found to be anaemic (PCV < 31%), with most of them displaying mild anaemia (95/99, 95.96%). Prevalence of anaemia was higher in males (54, 54.55%) than in females (45, 45.45%), although it was not statistically significant (χ^2^ = 0.020, *p* = 0.8886). The age group most affected was 5–10-year-olds (52, 52.53%). Anaemia prevalence was significantly higher in 5–10-year-old children compared with other age groups (χ^2^ = 9.590, *p* = 0.0083).

In this study, considering *Schistosoma* and STH as a risk factor for anaemia, there was a significant positive correlation between parasite-infected children and anaemia (χ^2^ = 38.093, *p* < 0.0001). The most prevalent parasite in the anaemic children group was *S. haematobium*, and there was a significant association between urinary schistosomiasis and anaemia (χ^2^ = 40.680, *p* < 0.0001).

### 3.3. Microscopic Examination

Overall, 172 (68.8%) participants were found to be negative for *Schistosoma* spp. and intestinal helminths by means of the methods used in the present study, and 78 (31.2%) participants were found to be positive for at least one of the parasites studied. Out of the 78 positive children, 41 (52.56%) were boys and 37 (47.44%) were girls.

Overall, 53 out of 78 (67.95%) of the children had a single infection and 25 out of 78 (32.05%) had more than one parasite (Table 1). The most frequent combinations for multiple parasitism can be visualized in Table 1. The maximum number of parasites found in the same child was three. All of the children with multiple parasites were boys (8–14 years), and the intensity of infection was light.

### 3.4. Prevalence and Intensity of Schistosomiasis Infection

The overall prevalence of urinary schistosomiasis among the children tested was 15.20%, 38 out of 250 (95% CI: 10.75%, 19.65%); 57.89% (22 out of 38) were boys and 42.11% (16 out of 38) were girls (Table 2). Most of them had a light infection with <50 eggs/10 mL of urine, and only in four out of 38 (10.53%) participants was the infection heavy (>50 eggs/10 mL of urine). Urogenital schistosomiasis was the most prevalent parasitic infection in the population studied, with a higher prevalence in children over 10 years (19.78%).

Regarding intestinal schistosomiasis, the overall prevalence was 8.00% (20 out of 250; 95% CI: 4.64%, 11.36%); 55% (11 out of 20) were boys and 45% (nine out of 20) were girls (Table 2). From the 20 KK positive cases, the epg range observed was from 48 to 2256, with a mean of 304.8 epg, giving in all cases moderate infection.

### 3.5. Prevalence and Intensity of STH Infection

The intestinal helminthic species identified and their prevalence, from highest to lowest, were *S. stercoralis* (10.80%; 27 cases; 95% CI: 6.95%, 14.65%), *A. lumbricoides* (7.20%; 18 cases; 95% CI: 4.00%, 10.40%), hookworm (1.20%, three cases; 95% CI: 0.00%, 2.55%), and *T. trichiura* (0.4%; one case; 95% CI: 0.000%, 1.182%) (Table 3). The gender-related prevalence of the intestinal helminths in the study area is shown in Table 2, and the age-related prevalence is summarized in Table 3. Among boys and girls, *S. stercoralis* revealed the highest difference, with 18 and nine positive cases, respectively, although the bivariate analysis did not confirm a significant association between gender and infection with *S. stercoralis*, (*p* = 0.1626; 95% CI, χ^2^: 1.955). Regarding the intensity of infection, A. lumbricoides had the statistically highest mean intensity (3958.67 epg), followed by *S. stercoralis* (387.56 larvae/g), *T. trichiura* (192 epg), and hookworm (64 epg).

The >10 years age group had the highest prevalence in any of the intestinal helminths. However, we cannot affirm that age and intestinal helminth infection are significatively associated.

## 4. Discussion

This study intended to determine the prevalence and intensity of urinary schistosomiasis and intestinal parasitic infections in school children in Osun State, Nigeria. Prior to this study, three similar reports existed on intestinal parasitic infections in the same State, but not on urinary schistosomiasis involving school children [15,16,17]. Ten years after Adefioye et al. and Sowemimo and Asaolu’s studies [15,17], whose authors reported an overall prevalence of 52.0% and 34.4%, respectively, our study reported an overall decreased prevalence of 27.60%, which is consistent with the rate of intestinal parasitic infection (24%) in a recent study conducted by Olopade et al., 2022 [16]. This indicates that in Osun State, the Federal Ministry of Health and Government of Nigeria’s interventions have made great achievements in the control of intestinal parasitic infections, and consequently there has been a prevalence reduction of these intestinal parasites [18]. In the past, the prevalence of STH in Osun State, Nigeria has been reported to be between moderate and high among children, and nowadays the prevalence is moderate [18].

Adefioye et al. and Olopade et al. recorded *A. lumbricoides* as the most prevalent intestinal parasite, at 36.2% and 22.1%, respectively, and the least prevalent parasites were *S. stercoralis* (0.7%) and *Hymenolepis nana* (0.3%), respectively [15,16]. In another study conducted by Aribodor et al. in Enugu State (Nigeria), they also found *A. lumbricoides* as the most prevalent STH, with a prevalence rate of 40.3%, followed by *T. trichiura* (15.3%) and hookworm (8.9%) [19]. Otherwise, in the present study, the most prevalent parasite was *S. stercoralis* (10.80%, 27 out of 250) and the least prevalent was *T. trichiura* (0.4%, one out of 250). Therefore, our study is not in congruence with most of the recent reports regarding intestinal parasites’ prevalence [15,16,19,20,21]. The true reason for these discrepant results in the area studied is not fully clear. It might have been due to the different diagnostic methods used, the amount of faeces utilized, or the number of stool samples used for the KK.

It should be noted that the KK technique presents a great variability in the considerable diagnostic results, due to the irregular distribution of eggs in the faeces depending on the samples. This explains the poorer performance of the KK technique and the low sensitivity of fecal examinations by microscopy [1,13,22,23].

In contrast, this study does agree with some previous reports in that none of the parasitic helminths were statistically gender-dependent or age-dependent, while this study also demonstrates that the parasitic intensity was light–moderate in all cases, except for *A. lumbricoides*, which had a high intensity [15,16,21].

Currently, the anthelminthic drugs recommended by the WHO for use in public health interventions to control STH infections are albendazole, levamisole, mebendazole and pyrantel [24]. The control of schistosomiasis and STH in Nigeria has employed preventive chemotherapy which involves the mass distribution of praziquantel and albendazole to school-aged children across endemic local government areas [18]. However, less than half (50%) of the treated endemic local government areas met the 75% effective coverage target in the last eight years. The unavailability of drugs and the logistics required to drive mass treatment campaigns are amongst the issues limiting coverage. These challenges were particularly worsened during the COVID-19 pandemic. Our study revealed the presence of schistosomiasis infection with a prevalence rate of 15.20% for *S. haematobium* and 8% for *S. mansoni*, indicating a moderate and low prevalence, respectively. Both *Schistosoma* species had, in general, light parasitic intensity, but unexpectedly 10.53% of the urinary schistosomiasis had a high infection rate (>50 eggs/10 mL of urine). These results are similar to others obtained in a study conducted in 2019, in which S. haematobium was detected in 13.6% of the students, while *S. mansoni* infection prevalence was 7.2% [19]. On the other hand, lower prevalence of *S. haematobium* (0.6%) and *S. mansoni* (2.3%) was recorded by Agbolade et al. compared to the present study [21,25]. Differences in geographical location, snail distribution, local endemicity of the parasites, and laboratory techniques used could explain the differing prevalence. The highest schistosomiasis prevalence was recorded among children over 10 years old compared to other age groups, but, as shown by this survey, age was not significantly associated with urinary and/or intestinal schistosomiasis.

The presence of human schistosomiasis is often related to sanitary deficiencies, contaminated water sources for domestic chores, bathing, and insufficient health education in the population or a deficiency in the control of the intermediate snail host. Temperature is also an important determinant of transmission of schistosomiasis, influencing parasite development and the lifecycle of snail intermediate hosts [1,13,23]. When praziquantel became available in the 1980s, given as oral tablets at a dose of 40 mg/kg body weight in Africa, MDA campaigns against schistosomiasis were slowly adopted as the major control strategy [26]. In Nigeria, MDA for schistosomiasis was first carried out in 2009 and aimed at reducing infection and morbidity [4,18]. The results presented here could be important not only to assess the control programs’ success and the prevalence reduction, but also to observe any changes in the incidence, distribution, and control of schistosomiasis disease and other factors relating to health resulting from schistosomiasis MDA.

The occurrence of anemia was statistically significantly associated with age. Children between five and 10 years old have a higher risk of developing anemia than other groups, and there is likely a considerable connection between anemia, intestinal helminth infections, and urinary schistosomiasis, with these parasites affecting hemoglobin levels in different ways, and this anemia is exacerbated when there is co-infection with *P. falciparum* [10,27,28]. Unfortunately, data regarding malaria parasites were not obtained in this study. This study shows that there was a significant association between anemia and helminth infection, especially with *S. haematobium*. A prevalence of 39.6% for anemia in our study reveals a worrying public health issue in Osun State. Fortunately, most of these cases were not severe anemia. Anemia is very common in developing countries, and it is generally a serious problem in school children [20].

## 5. Conclusions

In this study, it is observed that schistosomiasis and STH are still endemic in Osun State, but with a mild to moderate prevalence. Moreover, the intensity of infection has been found to be mild. Urinary infection was the most prevalent among the children recruited, with a higher prevalence in children over 10 years. The >10 years age group had the highest prevalence for all of the intestinal helminths. The association between gender and age and urogenital or intestinal parasites was not statistically significant.

## Figures and Tables

**Figure 1 diagnostics-13-00759-f001:**
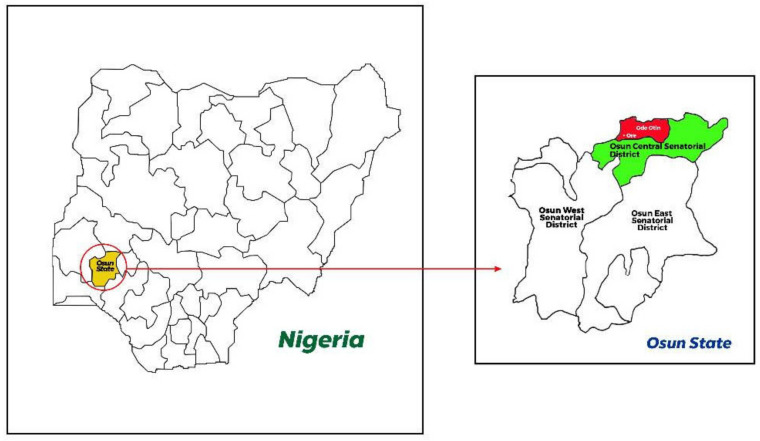
Map of Osun State (yellow in left map), Nigeria showing Ore community in Odo-Otin Local Government (red in right map) where the study was carried out. This community belongs to Osun Central Senatorial District (green in right map).

**Table 1 diagnostics-13-00759-t001:** Distribution of urogenital and intestinal schistosomiasis and intestinal helminths in the population studied and the most frequent combination of parasites.

	Parasites	Boys	Girls	Subtotal	Total
Single infection	SH	14	13	27	53
	SM	3	3	6	
	SS	6	5	11	
	AL	3	6	9	
	HOOKWORM	0	0	0	
	TT	0	0	0	
Total		26	27	53	
Multiple parasitism	AL + TT	0	1	1	25
	SM + SS + AL	2	0	2	
	SS + HOOKWORM	1	2	3	
	SM + SS	3	1	4	
	SS + AL	1	0	1	
	SM + AL	0	3	3	
	SH + SS	3	1	4	
	SH + SM	1	2	3	
	SH + SM + SS	2	0	2	
	SH + AL	2	0	2	
Total		15	10	25	78

SH: *Schistosoma haematobium*; SM: *Schistosoma mansoni*; SS: *Strongyloides stercoralis*; AL: *Ascaris lumbricoides*; TT: *Trichuris trichiura*.

**Table 2 diagnostics-13-00759-t002:** Total number of positive infections based on the species of parasite according to sex.

	SH		SM		SS		AL		HOOKWORM		TT		TOTAL
	P	N	P	N	P	N	P	N	P	N	P	N	N°
BOYS	22	113	11	124	18	117	8	127	1	134	0	135	**135**
GIRLS	16	99	9	106	9	106	10	105	2	113	1	114	**115**
TOTAL	38	212	20	230	27	223	18	232	3	247	1	249	**250**

P: Positive; N: Negative; SH: *Schistosoma haematobium*; SM: *Schistosoma mansoni*; SS: *Strongyloides stercoralis*; AL: *Ascaris lumbricoides*; TT: *Trichuris trichiura*.

**Table 3 diagnostics-13-00759-t003:** Urogenital and intestinal infections classified by age group.

Age Group	SH			SM			SS			AL			HOOKWORM			TT			TOTAL
	P	N	%	P	N	%	P	N	%	P	N	%	P	N	%	P	N	%	N°
<5 YEARS OLD	0	2	0.00	0	2	0.00	0	2	0.00	0	2	0.00	0	2	0.00	0	2	0	2
5–10 YEARS OLD	20	137	12.74	11	146	7.01	15	142	9.55	10	147	6.37	1	156	0.64	0	157	0.00	157
>10 YEARS OLD	18	73	19.78	9	82	9.89	12	79	13.19	8	83	8.79	2	89	2.20	1	90	1.10	91
TOTAL	38	212	15.20	20	230	8.00	27	223	10.80	18	232	7.20	3	247	1.20	1	249	0.4	250

P: Positive; N: Negative; SH: *Schistosoma haematobium*; SM: *Schistosoma mansoni*; SS: *Strongyloides stercoralis*; AL: *Ascaris lumbricoides*; TT: *Trichuris trichiura*.

## Data Availability

Not applicable.

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
