# Peer review of "Prevalence of Schistosoma haematobium and Intestinal Helminth Infections among Nigerian School Children"

_diagnostics, 2023, doi:10.3390/diagnostics13040759_

Round 1

Reviewer 1 Report

It is a status survey report of no originality. 

Author Response

Ref.:  Ms. No. Diagnostics-2151375

Prevalence of Schistosoma haematobium and intestinal helminths infections among Nigerian school children

Reviewer #1:

General comment

The present manuscript describes a small cross-sectional survey results of 250 school children by Schistosoma and intestinal helminths in Osun State, Nigeria. The area is a known endemic area of the parasitic helminths and all of the findings were similar with those of previous records. Therefore, the present study has little originality without new scientific advance. But it has value of record archiving of survey findings in that time and place.

Thank you very much for this comment. It is true that the study is not original in terms of methodology and the results were not surprising or bizarre either, but this updated prevalence is required for Osun State and the Federal Ministry of Health and the Government of Nigeria to further improve the control of these parasites in order to meet the 2030 goals.

Specific comments

  1. Explain how the 250 children were invited. Did you approach them underscientific way of sampling? Were the samples collected at schools that cooperated easily and voluntarily? How many schools?

Thanks for this comment. We have inserted a small paragraph including this information in the “2.3. Study population and sampling” section in the revised manuscript.

“Of the three schools in the community, only two participated in the study; the association of parents of the third school refused to participate.”

  1. Who did the microscopic exam? Introduce the microscopists.

Microscopy was performed in Nigeria by microscopists who are experts in detecting helminth eggs. This information has been added in the revised version.

“2.6 Microscopic examination

Microscopy was performed by a skilled, well-trained research microscopist who is an expert in detecting helminth eggs. 20% of prepared (previously examined) Kato-Katz slides were randomly selected for quality control and examined by a second experienced microscopist who was blinded to the previous test results.”

  1. If the findings of anaemia and parasitic infection were matched, the present study might make better scientific story. I strongly suggest to analyze the 99 anaemic children’s positivity of helminths.

Thanks for this valuable suggestion. Out of 99 anaemic children, 53 (53.54%) had some helminthic parasite, the rest of them 46 (46.46%) were not parasitized by any helminth. Analysing non-anaemic children (n=151), 126 (83.44%) had no parasites, and 25 (16.56%) had any kind of helminth infection. In statistical terms, this means that anaemia and helminth parasites are significantly associated(χ2= 38.093, P< 0.0001). Besides, S. haematobium is the parasite most prevalent in anaemic group and there is a significant association with anaemia (χ2= 40.680, P< 0.0001).

Given the value of these results, they have been included in the revised manuscript in the following sections.

3.2. Packed cell volume and anaemia

“In this study, considering Schistosoma and STH as a risk factor for anaemia, there was a significant positive correlation between parasite-infected children and anaemia (χ2= 38.093, P< 0.0001). The most prevalent parasite in the anaemic children group was S. haematobium, there was a significant association between urinary schistosomiasis and anaemia (χ2= 40.680, P< 0.0001).”

  1. Discussion

“As this study has shown there was a significant association between anaemia and helminth infection, especially with S. haematobium.”

  1. Table 2 and Figure 2 are duplicated. Table 2 with percentage records looks enough. The samples were not well organized and the age data (Figure 2b) have little significance.

Thank you for this appreciation. We also agree that Table 2 and Figure 2a have the same information, but are presented differently. Therefore, we decided to discard Figure 2a. Likewise, Figure 2b will not be presented in the revised manuscript because the information related to age and parasites is clearer in Table 3.

  1. Add one discussion paragraph of the low diagnostic sensitivity of the microscopy.

Thanks for this suggestion. With the reviewer's permission, we consider that this paragraph about microscopy and its disadvantages, in terms of specificity and sensitivity, is already known by all, and it is not necessary to repeat it in this study. Instead, we discussed in the original manuscript the Kato-Katz technique and its variability in results. We copy that paragraph here.

“It should be noted that the KK technique presents a great variability in the considerable diagnostic results, due to the irregular distribution of eggs in the faeces depending on the samples; this explains the poorer performance of the KK technique and the low sensitivity of faecal examinations by microscopy [1,13,22,23].”

  1. Add one paragraph introducing and discussing ongoing school deworming programs with albendazole/mebendazole and praziquantel in Nigeria.

We appreciate this suggestion. It really improves our manuscript. Thanks. We have added the following paragraph:

Currently, the recommended anthelminthic drugs by WHO for using in public health interventions to control STH infections are albendazole, levamisole, mebendazole and pyrantel [24]. The control of schistosomiasis and STH in Nigeria has been with preventive chemotherapy which involves mass distribution of praziquantel and albendazole to school-aged children across endemic local government areas [18]. However, less than half (50%) of the treated endemic local government areas met the 75% effective coverage in the last 8 years. The unavailability of drugs and logistics required to drive mass treatment campaigns are amongst the issues limiting coverage. These challenges were particularly worsened during the Covid-19 pandemic.”

Reviewer 2 Report

The study entitled “Prevalence of Schistosoma haematobium and intestinal helminths infections among Nigerian school children” by Alade et al. aims to investigate the prevalence of S. haematobium and STH infections in children of a specific endemic Nigerian community. Although an interesting study that, in general, can contribute to assess and maybe change the current control measures in place in the country (i.e., mass drug treatment), it has some limitations, which are mostly highlighted bellow.

1-     The English language must me improved throughout the whole manuscript.

2-     Abstract: needs to be improved and include basic description of items highlighted by the authors (e.g., levels that reflect light, moderate, high prevalence or intensity of infections. Please observe that S. mansoni causes the intestinal schistosomiasis but it is a blood parasite (fluke).

3-     Introduction:

L 46: please observe that preventive chemotherapy is different of mass drug treatment – which depend on previous assessment of the prevalence of the parasitic diseases in a determined region.

L 64-65: sentence must be removed or changed to the discussion.

L 74: the goal of the program is actually to control/eliminate schistosomiasis in the region through mass drug treatment. Please rephrase the sentence.  

L 84-89: the objectives must be better stated and some of them are never achieved in the study. Please revise.

4-     Methods:

Main concerns here are the number of samples and slides per samples (KK) analyses. I understand it was only one, which limits the power of detection and statistical analysis of the study.

Figure 1: please improve the legend and description. Be clear of what the highlighted parts of the maps represent.

L112-115: study design should already include the information exclusion criteria and when the participants where actually last treated.

The items should be re-organised in the sense they are confused and mix information. Item 2.4 should be before item 2.3 and item 2.5 should be part of the study population part. Information of samples collection, transport and storage must be included. Local of samples analyses as well.

Item 2/6. Include the anticoagulant (EDTA?).

L132 (item 2.7) : change the title: Urine filtration analysis.

L135: replace the word “pushed” – filtered.

L 138: delete “all the positive fields”.

L 142 (item 2.8): how many KK slides per sample were analyzed.

5-     Results

L 166-168:  sentence is misplaced – remove it.

L 177:  must be corrected  …were found negative for S. haematobium and SH by the  methods used in the present study”.

L178: Positive for.

L182  & throughout the manuscript and tables: italicize the scientific names.

L195: Delete “surprisingly”.

Table 1- improve heading

L196: remove “urinary infection”. Use urogenital schistosomiasis was the most prevalent parasitic infection in the population studied.

L200: I am confused of how the authors arrived in an epg range of 2-94 when if only a single egg in KK should result in 24 epg. Please explain this result and correct the results accordingly.

 Table 2: confusing and data for STH unnecessary as it repeats information already present in Table 1.

Figure 2: please improve legend and description.

 In general, presentation of the results must be improved, and item 3.3 is unnecessary and makes the whole thing repetitive. The Tables must be adjusted and probably only one or two of them will suffice.

6-     Discussion:

Improve to include more about the MDT and impact of Praziquantel and Mebendazole treatment in the prevalence found.

Author Response

Thank you very much for your comments. Reply to the Review Report has been uploaded.

Round 2

Reviewer 1 Report

Most of the concerns have been addressed well.